# Imaging and Tissue Biomarkers of Choline Metabolism in Diffuse Adult Glioma: 18F-Fluoromethylcholine PET/CT, Magnetic Resonance Spectroscopy, and Choline Kinase α

**DOI:** 10.3390/cancers11121969

**Published:** 2019-12-07

**Authors:** Matthew Grech-Sollars, Katherine L Ordidge, Babar Vaqas, Claire Davies, Vijay Vaja, Lesley Honeyfield, Sophie Camp, David Towey, Helen Mayers, David Peterson, Kevin O’Neill, Federico Roncaroli, Tara D Barwick, Adam D Waldman

**Affiliations:** 1Department of Surgery and Cancer, Imperial College London, London SW7 2AZ, UK; m.grech-sollars@imperial.ac.uk (M.G.-S.); claire_m_davies@yahoo.com.au (C.D.);; 2Department of Imaging, Imperial College Healthcare NHS Trust, London SW7 2AZ, UKlesley.honeyfield@nhs.net (L.H.); 3Department of Neurosurgery, Imperial College Healthcare NHS Trust, London SW7 2AZ, UK; b.vaqas@nhs.net (B.V.); sophie.camp@nhs.net (S.C.); david.peterson@nhs.net (D.P.); kevin.oneill@nhs.net (K.O.); 4Department of Brain Sciences, Imperial College London, London SW7 2AZ, UK; v.vaja@imperial.ac.uk; 5Northampton General Hospital NHS Trust, Northampton NN1 5BD, UK; david.towey@ngh.nhs.uk; 6Department of Cellular Pathology, Salford Royal Foundation Trust, Salford M6 8HD, UK; helen.mayers@srft.nhs.uk; 7Division of Neuroscience and Experimental Psychology, University of Manchester, Manchester M13 9PL, UK; federico.roncaroli@manchester.ac.uk; 8Centre for Clinical Brain Sciences, The University of Edinburgh, Edinburgh EH8 9YL, UK

**Keywords:** brain, choline, glioma, MRI, PET

## Abstract

The cellular and molecular basis of choline uptake on PET imaging and MRS-visible choline-containing compounds is not well understood. Choline kinase alpha (ChoKα) is an enzyme that phosphorylates choline, an essential step in membrane synthesis. We investigate choline metabolism through 18F-fluoromethylcholine (18F-FMC) PET, MRS, and tissue ChoKα in human glioma. Fourteen patients with a suspected diffuse glioma underwent multimodal 3T MRI and dynamic 18F-FMC PET/CT prior to surgery. Co-registered PET and MRI data were used to target biopsies to regions of high and low choline signal, and immunohistochemistry for ChoKα expression was performed. The 18F-FMC/PET differentiated WHO (World Health Organization) grade IV from grade II and III tumours, whereas MRS differentiated grade III/IV from grade II tumours. Tumoural 18F-FMC/PET uptake was higher than in normal-appearing white matter across all grades and markedly elevated within regions of contrast enhancement. The 18F-FMC/PET correlated weakly with MRS Cho ratios. ChoKα expression on IHC was negative or weak in all but one glioblastoma sample, and did not correlate with tumour grade or imaging choline markers. MRS and 18F-FMC/PET provide complimentary information on glioma choline metabolism. Tracer uptake is, however, potentially confounded by blood–brain barrier permeability. ChoKα overexpression does not appear to be a common feature in diffuse glioma.

## 1. Introduction

Gliomas are the most common primary brain tumour in adults and one of the most frequent cause of cancer related death in young adults aged 20–39 in countries with a very high human development index [1]. Behaviour and prognosis of individual gliomas is highly variable and there is a need for objective imaging biomarkers to permit reliable stratification for appropriate therapy. MRI is the modality of choice for the diagnosis and follow-up of patients with a glioma. Advanced MR techniques are becoming more widely used to characterize altered metabolism and other pathological features in these tumours. Choline is a naturally existing substrate that is transported intracellularly and phosphorylated in cells to phosphocholine by the enzyme choline kinase alpha (ChoKα), to eventually yield phosphatidylcholine, an essential phospholipid for cell membrane production [2]. Deregulated cell proliferation in malignant cells is associated with increased membrane turnover and higher concentrations of membrane components manufactured via the Kennedy pathway [3]. Thus, choline metabolism is recognized as a marker of tumour cell proliferation [4]. 

Magnetic resonance spectroscopy (MRS) can measure tumour metabolites non-invasively. The Cho resonance at 3.2 ppm, comprises varying proportions of ‘choline-containing compounds’ (predominantly phosphocholine, phosphoethanolamine, glycerophosphocholine, and glycerophosphoethanolamine) involved in phospholipid synthesis and degradation [3]. Cho is often expressed as a ratio to creatine/phosphocreatine (Cr, 3.0 ppm). Several studies in glioma have shown a correlation between elevated MRS Cho levels and the cell proliferation marker Ki-67 [5,6], as well as correlation of the Cho/Cr ratio with Ki-67 [7,8]. The choline signal on MRS is thought to be useful in determining the grade of glioma [9] and the presence of anaplastic transformation [10].

Choline metabolism may be evaluated non-invasively in-vivo using radiotracers imaged with PET. 18F-fluoromethylcholine (18F-FMC) has been shown to mimic closely endogenous choline uptake in normal tissue and prostate cancer [11]. In glioma imaging, pilot clinical studies have shown differential uptake of choline radiotracers between glioma and normal brain tissue, and between glioma and other disease processes. For example, a pilot study in 12 patients by Hara et al. found higher tumour-to-background tissue ratios using 18F-fluoroethylcholine compared with 11C-choline [12]. Subsequently, a larger study of 30 patients looking at 18F-FMC uptake in solitary brain lesions showed differential uptake between benign lesions, high grade gliomas, and metastatic lesions [13]. The relationship between choline radiotracer uptake and MRS-visible metabolites has not previously been studied in human glioma.

ChoKα is known to be overexpressed in lung, prostate, and colorectal carcinomas [14]. In ex-vivo glioma cells, genomic analysis has linked phosphocholine measured on MRS to the expression of choline kinase β (ChoKβ) and phospholipase genes [15]. However, in this small study, no increased expression of ChoKα was identified. ChoKα is, however, considered to be essential for phosphatidylcholine synthesis, and in vitro tumour cell line studies have shown overexpression of ChoKα and not ChoKβ [16]. How ChoKα expression is related to imaging markers of choline metabolism in vivo in glioma is unclear.

In this prospective study, we investigated the relationship between choline metabolites measured with MRS, uptake of 18F-FMC with PET/CT imaging, and expression of ChoKα in spatially correlated glioma tissue from targeted biopsies.

## 2. Method

### 2.1. Patients

Fourteen patients (7 male, 7 female; aged 23–73 years, mean 40 years) were recruited to this study. Inclusion criteria were age >18 years, suspected primary supratentorial glial tumours, lesions greater than 2 cm, WHO (World Health Organization) performance status ≤2, considered suitable for biopsy and/or resection by the neuro-oncology MDT at Imperial College Healthcare NHS Trust. One patient was diagnosed with a dysembryoplastic neuroepithelial tumour (DNT) and was excluded from further tissue analysis. Patients previously treated with radiotherapy, cytotoxic agents, or other experimental drugs were excluded from the study. The study was reviewed and approved by the London—Fulham Research Ethics Committee. Written and informed consent was obtained before recruitment to the study and all data were anonymized in accordance with the General Data Protection Regulation (GDPR). 

### 2.2. Image Acquisition

PET/CT (13/14 patients) and MRI (14/14 patients) were performed prior to surgery; both modalities on the same day in 12/14 patients, and in one case 7 days apart. 

MR images were acquired on a 3T Siemens Verio MRI system (Siemens, Erlangen, De; VB19) using a 32-channel head coil. The MR protocol included pre- and post-gadolinium high resolution 3D volumetric T1 images (voxel size 1 × 1 × 1 mm); MR spectroscopy (MRS) using a 2D chemical shift imaging (CSI) sequence (PRESS: TE = 30 ms, TR = 1700 ms, voxel size 10 × 10 × 15 mm); and total acquisition time for one CSI slice was 7 min. Between one and three CSI slices were acquired per patient to cover the whole tumour where possible. 

PET/CT was performed on a Siemens Biograph 6 scanner. Following patient positioning, a CT scan (50 mAs, 110 kV, 0.6 pitch, 1.0 sec rotation, 6 × 2.0 mm collimation, 3 mm slices) was performed for attenuation correction and co-registration. The 18F-FMC (PETNET Solutions Inc., Northwood, UK) was injected as a bolus injection intravenously (mean activity 261 MBq, target maximum activity 285 MBq) and a dedicated 45-min brain dynamic list mode acquisition was performed. The administration of radioactivity for the PET scans was approved by the Administration of Radioactive Substances Advisory Committee, United Kingdom. In one patient, 154 MBq of 18F-FMC was injected, and in another patient the acquisition was halted after 38 min as the patient needed to micturate. 

Discrete venous blood samples of 5 mL were taken during scanning in 8 of the 14 patients. Samples were taken at 2.5, 5, 7.5, 10, and 45 min post-injection in the first two patients and at 1, 2, 3, 4, 5, 10, and 45 min post-injection in the next 6 patients. The start and end withdrawal times for the venous sampling were recorded.

### 2.3. Tumour Biopsies

Thirteen of the 14 patients underwent surgical resection or biopsy at Charing Cross Hospital, London UK, 1–51 days (mean 17 days) after imaging. One patient opted to undergo surgical resection at a different institution, precluding ChoKα analysis, although neuropathological diagnosis was made available for that patient. Regions of both high and low Cho/Cr ratios on MRS and 18F-FMC uptake on PET were marked for biopsy by neurosurgeons using color-coded hollow virtual spheres represented in the surgical neuronavigational system using methodology described previously [17]. Biopsies were obtained in 1–4 regions per patient and the regions sampled were recorded on the neuronavigational systems for correlation with imaging parameters. One to two samples from each patient were used for ChoKα analysis.

### 2.4. Tissue Analyses

Samples were analysed at The University of Manchester and Imperial College London using both automated and manual immunohistochemistry staining. Tumour specimens were evaluated by a neuropathologist (F.R.) according to the integrated approach recommended by the revised 4th edition of the WHO classification of brain tumours [18]. IHC for ChoKα was performed on 1–2 samples per patient to cover a range of high to low Cho/Cr ratios on MRS and high to low 18F-FMC uptake on PET. Serial 4 µm sections from formalin fixed, paraffin embedded tissue were cut on to Superfrost Plus slides then heated at 60 °C for 45 min prior to staining.

IHC ChoKα detection was performed both by manual and automated methods using two commercially available ChoKα antibodies as described below. An appropriate positive control (bronchial biopsy) was included for each slide. A negative control was performed by the addition of diluent only during primary antibody incubation. IHC stained sections were reviewed by a neuropathologist (F.R.).

Manual staining of clone HPA024153 (Sigma-Aldrich, St. Louis, MO, USA) was performed as follows. Slides were dewaxed by immersion in Histo-Clear (National Diagnostics USA, Atlanta, GA, USA) and rehydrated with subsequent immersion in 100% ethanol, 70% ethanol, and distilled water. Antigen retrieval was performed by immersion in 95 °C TRIS-EDTA buffer (Sigma-Aldrich Ltd.) comprised of 1M Tris-HCl (pH approximately 8.0) containing 0.1M EDTA for 30 min in a Grant SUB Aqua 5 Plus water-bath. Slides were rinsed in PBS and endogenous peroxidase activity blocked by immersion in 0.3% hydrogen peroxide (Sigma-Aldrich Ltd.) in PBS for 15 min. Thereafter, slides were rinsed with PBS and incubated with 1.5% normal goat serum (Vector Laboratories, Burlingame, CA, USA) for 30 min prior to incubation with primary antibody HPA024153 (Sigma-Aldrich, 1/80) at room temperature for one hour. Slides were rinsed in PBS and incubated with secondary biotinylated antibody (Goat Anti-Rabbit IgG, Vector Laboratories, 1/100) for 30 min followed by an avidin/biotin peroxidase complex (VECTASTAIN Elite ABC Kit, Vector Laboratories) for 30 min. Chromogenic reaction was developed using DAB (Diaminobenzidine, Vector ImmPACT DAB Peroxidase Substrate) for 1 min then halted by immersion in running tap water for 5 min. Nuclei were counterstained with Gill 2 Haematoxylin (Thermo Fisher Scientific Shandon, Runcorn, UK) and blued in Scott’s tap water (in-house preparation) for 1 min. Slides were dehydrated in 70% ethanol and 100% ethanol, cleared in Histo-Clear (National Diagnostics USA, Atlanta, GA, USA), and mounted in DPX (VWR BDH ProLab, Wirral, UK). 

Automated staining of clone NBP1-85630 (Novus Biologicals, Littleton, CO, USA) was performed using a Ventana Roche Benchmark Ultra autostainer (Hoffmann-La Roche LTD, Atlanta, CA, USA). Slides were subjected to 76 min heat-induced epitope retrieval using Ventana Ultra CC1 (EDTA based pH 9.0) prior to primary antibody incubation at 1/100 for 60 min. Detection and subsequent nuclear counterstaining were performed with Ventana Ultraview Detection Kit and Ventana Haematoxylin. Slides were dehydrated in gradated ethanol, cleared in xylene, and mounted in DPX (VWR BDH ProLab, Wirral, UK).

### 2.5. Data Analyses

#### 2.5.1. Venous Sampling

Start and end withdrawal times were recorded for each sample taken from the venous cannula, and the mean withdrawal time calculated. Well-counting of blood samples was carried out for each time-point to quantify the level of radioactivity in each sample, followed by well-counting of plasma obtained from spun-down blood samples. Venous blood samples were processed before high-performance liquid chromatography (HPLC) analysis. Proteins were removed using Acetonitrile precipitation of cell free plasma, followed by vacuum rotary evaporation. The dried-down sample was reconstituted with mobile phase followed by Millipore filtration. The filtered solution was then injected onto a 1 mL sample loop, in readiness for HPLC analysis. The resultant solution was analysed with sensitive radioactivity detection, by HPLC (Phenonenex Luna 10 μ SCX 100 A, 250 × 4.6 mm i.d.) at a flow rate of 2 mL/min. The mobile phase used for elution was sodium di-hydrogen orthophosphate (0.25 M) and acetonitrile (90:10). The radioactive detector was linked to a PC based integrator, Agilent software, and the 18F-Choline content obtained from integrated radioactivity traces. The fraction of parent compound was measured and plotted against the mean withdrawal time. 

#### 2.5.2. Dynamic Image Analysis

Raw PET data were attenuation and scatter corrected, then reconstructed iteratively (dynamic data: 2 iterations, 8 subsets; static data: 8 iterations, 16 subsets). Processed PET data were analysed using a dedicated PET workstation (Hermes Medical Solutions, Stockholm, Sweden). The dynamic PET images were reviewed alongside MR images by two experienced observers (T.B., A.W.) in consensus to manually outline volumes of interest (VOI) for quantitative standardised uptake value (SUV) analysis and to generate time activity curves. Dynamic data was available in 12/13 patients who had a PET scan. One of the patients was excluded from the dynamic data analysis due to motion artefact. In these 12 patients, time activity curves of *SUV_max_* against time were drawn for each of the tumours according to WHO grade and for a 2 cm spherical region of interest (ROI) in contralateral white matter. 

#### 2.5.3. Static Image Analysis

Considering the time activity curves and results from the venous sampling, the most stable time period of 7–17 min was chosen as the optimum time window for reconstruction and analysis of static 18F-FMC PET images in all 13 patients who underwent PET. These images were used for analysis of *SUV_max_* and measurements of the tumour-to-background ratio (TBR). TBR was calculated considering the tumour *SUV_max_* and *SUV_mean_* in a 2 cm diameter spherical ROI in the contralateral white matter (WM) as shown in Equation 1 and presented in [13,19].

(1)$$\mathrm{TBR}=\frac{\left[{SUV}_{max}\right]\mathrm{tumour}}{\left[{SUV}_{mean}\right]\mathrm{WM}}$$

#### 2.5.4. MR Spectroscopy

MRS data were analysed using TARQUIN [20] to extract values for total choline containing compounds (Cho) and total creatine containing compounds (Cr). Data were eddy current corrected and the ratio of Cho/Cr was extracted for all relevant voxels. The TARQUIN Cho/Cr ratio was converted from relative metabolite concentration to peak intensity by multiplying it by 9/3 for comparison with literature values. [Cho/Cr]*_max_* was measured as the maximum Cho/Cr ratio within the tumour regions across all slices over which the spectroscopy data were acquired. Relative values were used to select high and low Cho/Cr regions within individual tumours. A pragmatic Cho/Cr threshold of 2.4 was used to define regions of high and low Cho/Cr for the purposes of comparison with ChoKα staining.

#### 2.5.5. Imaging vs. Tumour Grade

SUV_max_, TBR, and [Cho/Cr]*_max_* ratios on MRS were used to interrogate the ability of these imaging parameters to differentiate between the various tumour grades. The Shapiro–Wilk test was used to test for normality. As the distributions were found to be not normal, the Wilcoxon rank-sum test was used to test the difference in means between the various tumour grades. Data were tested over 12 patients for PET and 13 patients for MRS: Patient 13 (P013) was excluded from analyses as they were diagnosed as a diffuse neuroepithelial tumour (DNET), and for patient 6 (P006), PET data were not acquired. 

#### 2.5.6. Correlating 18F-FMC PET Uptake and Cho/Cr Ratio

The Cho/Cr ratio on MRS and the TBR on 18F-FMC PET were correlated to analyse the relationship between the two imaging parameters, both in terms of spatial location as well as the numerical measure. MRS location was extracted and compared to the PET images as previously described [17]. PET static images were registered to the post-contrast T1 weighted image over which the MRS voxel locations were extracted using a normalized mutual information method and trilinear interpolation in FLIRT (FSL). The co-registered PET image, post-contrast T1 weighted image, and the MRS voxel locations were overlaid in order to compare the spatial location differences for [Cho/Cr]*_max_* and TBR. The quantitative values were then correlated by extracting the SUV values in each region of viable MRS data. The sample to background ratio (SBR) was measured as the mean SUV in the MRS ROI divided by the mean SUV in the contralateral white matter. The SBR was plotted against the Cho/Cr ratio, grouping patients into the different tumour grades. The correlation between the two was tested using the Pearson correlation coefficient in MATLAB.

(2)$${\mathrm{SBR}}_{\mathrm{ROI}}=\frac{\left[{SUV}_{mean}\right]\mathrm{ROI}}{\left[{SUV}_{mean}\right]\mathrm{WM}}$$

#### 2.5.7. Choline Uptake and Contrast Enhancement on MRI

Regions of very high 18F-FMC uptake were also correlated with regions of contrast enhancement by overlaying the two and measuring the percentage overlap between contrast enhancement and high 18F-FMC uptake. ROIs were drawn on contrast enhancing regions for each of the enhancing tumours. Regions with the highest 18F-FMC uptake within the tumour were defined as at least three times the *SUV_mean_* of the contralateral white matter. The percentage of contrast enhancing voxels with high 18F-FMC uptake was then calculated by dividing the number of voxels with high 18F-FMC uptake within the contrast enhancing region over the number of voxels within the contrast enhancing region. Non contrast enhancing regions of tumours that were hyperintense on T2 FLAIR were also defined, and the percentage of voxels with an uptake higher than the threshold was also calculated for comparison.

#### 2.5.8. Correlation between Imaging and Tissue Parameters

Spherical ROIs (8 mm diameter) were drawn using MRIcron in each of the biopsied regions using recordings obtained from the neuronavigational software. In addition, a 2 cm diameter spherical ROI was drawn around contralateral white matter for each patient to measure the sample-to-background ratio (SBR_biopsy_) for each biopsied sample. The ROIs were drawn on the post-contrast T1 weighted image, to which the static PET images were registered. A selection of 1 to 2 samples per patient were chosen to cover a range of SBR_biopsy_ and [Cho/Cr]_biopsy_ ratios for further tissue analyses. 

## 3. Results

### 3.1. Neuropathology

A summary of patient tissue data, including molecular diagnosis, is shown in Table 1. Of the 14 patients, 3 were diagnosed as WHO grade IV tumours, 4 as WHO grade III tumours, 6 as WHO grade II tumours, and 1 as a WHO grade I tumour. The patient diagnosed with a dysembryoplastic neuroepithelial tumour was excluded from comparative analyses. 

### 3.2. Metabolite Analysis

Results for the metabolite analysis carried out through venous sampling in 8 patients are shown in Figure 1A. Fifty percent of the 18F-FMC was metabolized between 7.5 min (WHO grade II tumour) and 42.5 min (WHO grade III tumour) post-injection. The main metabolite was betaine. A second and third metabolite peak were observed in three patients at a single time-point. Following results from the first two patients, which showed rapid metabolism of the parent compound, venous sampling was changed from every 2.5 min in the first 10 min to every minute in the first five minutes.

### 3.3. Time Activity Curves 

Time activity curves (Figure 1B) show rapid uptake within the first five minutes and increased uptake in all tumours compared to contralateral white matter. Considering the time activity curves and metabolite analysis for each of the individual patients, the 7–17 min time period was identified as the most stable for static image analysis. 

### 3.4. TBR and *SUV_max_* vs. Tumour Grade

Using both TBR and *SUV_max_* we were able to differentiate between WHO grade IV tumours and the rest of the tumours combined (*p* < 0.01). A box plot for *SUV_max_* by tumour grade is shown in Figure 2A. The highest uptake was observed in WHO grade IV tumours. All tumours had a higher uptake than that of the respective normal-appearing contralateral white matter. 

### 3.5. MR Spectroscopy vs. Tumour Grade

Results for comparing maximum choline to creatine ratios according to tumour grade are shown in the boxplot in Figure 2B. Using [Cho/Cr]*_max_* we were able to differentiate between WHO grade II and WHO grade III tumours (*p* < 0.01). Grouping WHO grade III and WHO grade IV tumours as high-grade tumours, and WHO grade II tumours as low-grade tumours, showed a statistically significant difference between the two groups (*p* < 0.01).

### 3.6. FMC-PET Uptake vs. MR Spectroscopy

Of the 14 patients, 1 did not have good quality spectra, 1 did not have PET images, and 3 did not have significantly elevated 18F-FMC uptake. In the remaining 9 patients, regions of highest 18F-FMC uptake correlated spatially with regions of highest Cho/Cr ratio in the regions where good quality MRS data were available. 

The correlation between 18F-FMC-uptake and Cho/Cr ratio was tested in 12/14 patients and 257 MRS voxels were used. Correlating the two measures showed a significant moderate-positive correlation between 18F-FMC uptake and Cho/Cr ratio (r = 0.59, *p* << 0.001). When considering WHO grade IV tumours alone, a strong positive correlation between the two measures (r = 0.9, *p* << 0.001) was demonstrated. It was noted that all of the data points shown in Figure 3 with an SBR over 10 resulted from a single patient. Excluding that patient from the analysis showed a weak positive correlation between SBR and Cho/Cr ratio (r = 0.24, *p* << 0.001). 

### 3.7. ChoKα Expression

Optimization for ChoKα staining in testes is shown in Figure 4A,D; results are similar to those shown on Protein Atlas [21]. Tissue sample details are shown in Table 1. Of the 22 samples analysed, only one showed strong ChoKα staining. Samples with high PET uptake (defined as regions with an SBR greater than 2.0) and high Cho/Cr on MRS (defined as a Cho/Cr greater than 2.4) did not correlate with tissue expression of ChoKα. Figure 4 shows sample images from regions with both high and low PET uptake, exhibiting both positive and negative ChoKα. Similar results were obtained for MRS. Of note, staining for ChoKα was only strong in one of the 22 samples (compare Figure 4A–C). This sample was from a patient with GBM with the highest choline uptake on PET and second highest Cho/Cr on MRS (Figure 4B). However, a second sample from this patient in a region of high choline uptake and Cho/Cr was negative for ChoKα. 

### 3.8. Choline PET Uptake vs. Contrast Enhanced T1 MRI

Seven of the patients showed regions of contrast enhancement (3 WHO grade IV, 2 WHO grade III, 1 WHO grade II, 1 WHO grade I). Overlaying areas of highest 18F-FMC uptake on contrast-enhanced T1 weighted imaging (Figure 5) showed that on average 94% of the volume of contrast enhancement had an uptake above the threshold, which was defined as three times the uptake in contralateral white matter. Of the 7 patients showing regions of contrast enhancement, 5 showed high 18F-FMC uptake in 100% of the contrast enhancing regions, 1 in 92% (WHO grade IV), and 1 in 67% (WHO grade III). In comparison, when analysing hyperintense regions on T2 FLAIR and excluding regions of contrast enhancement in these 7 patients, an average of 39% (min 1%, max 89%) of the volume of non contrast enhancing tumour showed 18F-FMC uptake greater than the threshold. Of note, the grade I DNET, which was excluded from formal comparative analysis, showed contrast enhancement and markedly elevated 18F-FMC uptake.

## 4. Discussion

### 4.1. 18F-FMC Metabolism

The 18F-FMC PET was metabolized rapidly. Between 46% and 72% of the 18F-FMC was metabolized at 10 min post injection. When considering late PET image analyses, the majority of the radiotracer would have been metabolized to betaine, making it difficult to discriminate between the contributions of the parent compound and the catabolite [22]. The time activity curves showed rapid uptake of choline in the tumours within the first five minutes and relatively stable activity thereafter, in keeping with the findings of Mertens et al. [19]. Based on the TAC and metabolite analysis, 7–17 min was chosen as the most stable time point for static image analyses of PET data. 

### 4.2. 18F-FMC Uptake and Cho/Cr in Glioma Grading

The 18F-FMC PET and MRS appear to give complimentary information with regard to glioma grade. The 18F-FMC PET uptake differentiated glioblastomas (WHO IV) from grade II/III, whereas Cho/Cr distinguished between grade II and grade III/IV lesions, which is diagnostically important in terms of current treatment regimens. Numerous previous studies have shown the potential of MRS, which has limited spatial resolution, to distinguish between low and high grade tumours [9,10]. In our study, high 18F-FMC uptake appears to identify GBM. An earlier study of 11C-choline and 18F-FMC in 12 suspected glioma cases also showed higher choline uptake in higher grade tumours, but their study comprised predominantly WHO Grade IV cases [12]. The degree to which uptake in GBM is confounded by the blood–brain barrier breakdown in these enhancing tumours is unclear. 

Molecular subtypes, particularly IDH and 1p/19q, as well as grade are important determinants of glioma phenotype and behaviour as emphasized in the WHO 2016 guidelines, and further work is required in a larger cohort of patients to explore relationships between these and imaging markers of choline metabolism. Importantly, both 18F-FMC PET and MRS offer spatial information which can be used to distinguish regions that are more aggressive to guide targeted representative biopsies. This is well-illustrated in one patient with anaplastic oligodendroglioma, where regions of high choline uptake on PET correlated with the histologically higher-grade component of the tumour.

### 4.3. 18F-FMC PET Uptake and Contrast Enhancement

High 18F-FMC uptake was observed in 94% of contrast enhancing regions in seven patients. Although this may reflect high localized metabolic activity, 18F-FMC uptake may be related to blood–brain barrier deficiency. Structures that lack a blood-brain barrier (e.g., choroid plexus) have been shown to display higher choline uptake than other regions of the brain [19]. Furthermore, a more recent review indicated that choline radiotracers are affected by blood–brain barrier damage [23]. Our finding of high uptake in a DNET, a tumour type known to have low metabolic activity and negligible malignant potential but in this case avid contrast enhancement, supports the view that this may be a dominant mechanism. Modestly elevated choline uptake relative to normal-appearing contralateral white matter in non-enhancing tumours does, however, suggest that some uptake is genuinely due to choline metabolism. Further kinetic studies to examine the effects of blood–brain barrier permeability on 18F-FMC uptake are needed to better understand 18F-FMC as a probe of glioma choline metabolism and its clinical utility.

### 4.4. 18F-FMC PET vs. MRS 

We found a moderate and significant positive correlation between SBR and the respective Cho/Cr ratio over the whole tumour across all patients. Although there is a relationship between these two parameters, there are clear differences in the way the modalities differentiate between tumour grade. The processes probed by the two modalities differ: MRS Cho signal depends on steady-state levels of intracellular compounds involved in both membrane synthesis and breakdown, whereas 18F-FMC PET detects carrier-mediated flux of choline as a substrate for membrane synthetic pathways. Moreover, the latter may also be confounded by non-specific uptake related to blood–brain barrier disruption, as discussed above. Further studies are required to confirm the dominant processes influencing these choline-related signals, and how they relate to key pathological processes in glioma.

### 4.5. Choline Kinase Alpha Expression

The largely negative results for ChoKα staining in the tissue samples were unexpected, and explain the absence of significant correlation with imaging markers of choline metabolism, tumour grade, and phenotype. We have not identified the significant levels of ChoKα expression seen in isolated cell lines and correlated with 11C-choline uptake in vivo in other tumour types. In our cohort, it is unlikely that the weak ChoKα staining observed in all but one of the positive samples was sufficient to explain the 18F-FMC uptake on PET and elevated Cho/Cr on MRS. The degree to which xenografts and syngeneic pre-clinical models recapitulate the metabolic environment in human diffuse gliomas, especially those with IDH and p53 or 1p/19q mutations in unclear. High levels of staining were present in only one sample from an IDH wild type GBM, however, another sample from the same tumour and other wild type (WT) gliomas showed negative staining. We targeted ChoKα expression as the most promising candidate tissue marker; it is possible that the ChoKβ isoform or other synthetic pathways may be of more relevance in gliomas. Phospholipid synthesis pathways, and in particular ChoKα, have been proposed as potential therapeutic targets, and our negative results may be important in informing such treatment strategies. In addition, the contribution of non-tumour cells such as infiltrating immune cells could be addressed in future studies by labelling for microglia/macrophages.

### 4.6. Limitations

The study was limited by the small number of patients. In particular, there were only three patients with GBM, and only three IDH wild type gliomas, although multiple samples were taken from each lesion. Limitations for the correlation between imaging and tissue biomarkers include the limited spatial resolution of the imaging techniques, relative imaging parameters applied for selecting biopsy location, and accuracy of the neuronavigational system; we used a region with an 8 mm diameter, allowing for an error margin that could be related to the system used or brain shift as discussed in [17]. Given the low ChoKα expression in this series, this is unlikely to have significantly biased our results. The rapid metabolism of 18F-FMC to betaine also results in signals from a mixture of chemical species in the later phases of PET acquisition. For practical reasons we have limited tissue analysis to ChoKα, although, as discussed above, other enzymatic markers bear further exploration in this tumour type.

## 5. Conclusions

Correlation with relevant tissue features is important for better understanding and validating of imaging biomarkers in spatially heterogenous diseases. ChoKα expression in diffuse glioma, however, appears generally low, which limits the mechanistic conclusions that can be drawn in this instance. The 18F-FMC PET and MRS provide complimentary information on glioma grade, and can facilitate targeted biopsies from the most aggressive regions in order to guide individual patient management. Improved understanding of factors affecting choline tracer uptake are needed for widespread clinical implementation. 

## Figures and Tables

**Figure 1 cancers-11-01969-f001:**
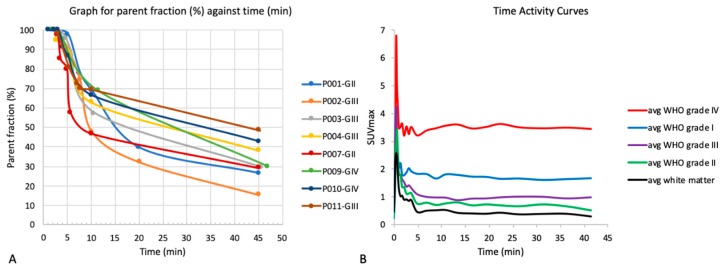
(**A**) Metabolite analysis. The graph shows a plot of the parent fraction against the mean withdrawal time for venous sampling from 8 patients. The time taken to metabolize 50% of 18F-FMC was 7.5–42.5 min. (**B**) Time activity curves by tumour grade. The plot of maximum standardised uptake value (*SUV_max_*) against time is shown for each of the tumour grades grouped together and the average contralateral white matter. Uptake was higher in all tumours compared to contralateral white matter with the highest uptake in the WHO grade IV tumours. Uptake of 18F-FMC was rapid within the first 5 min before reaching a steady-state.

**Figure 2 cancers-11-01969-f002:**
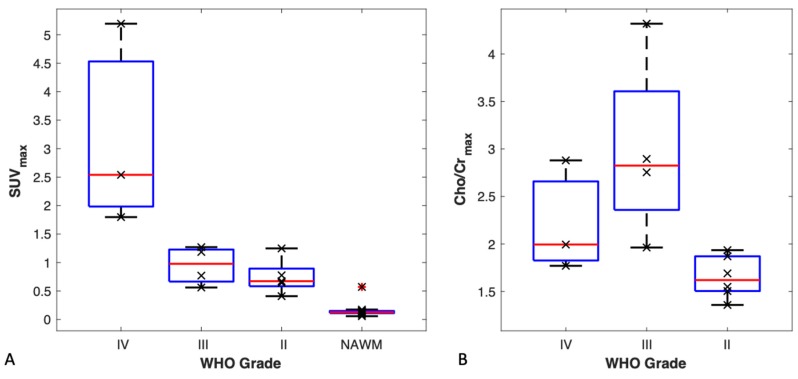
(**A**) *SUV_max_* by tumour grade. Box plots for *SUV_max_* are shown by tumour grade and in normal appearing contralateral white matter (NAWM). Wilcoxon rank-sum test showed a significant difference between WHO grade IV tumours and all other tumours. *SUV_max_* was unable to differentiate between the other tumours. (**B**) Maximum Cho/Cr ratio by tumour grade. Box plots for [Cho/Cr]*_max_* by tumour grade show a statistically significant difference between WHO grade III and WHO grade II tumours but not between grade IV tumours and the other tumours.

**Figure 3 cancers-11-01969-f003:**
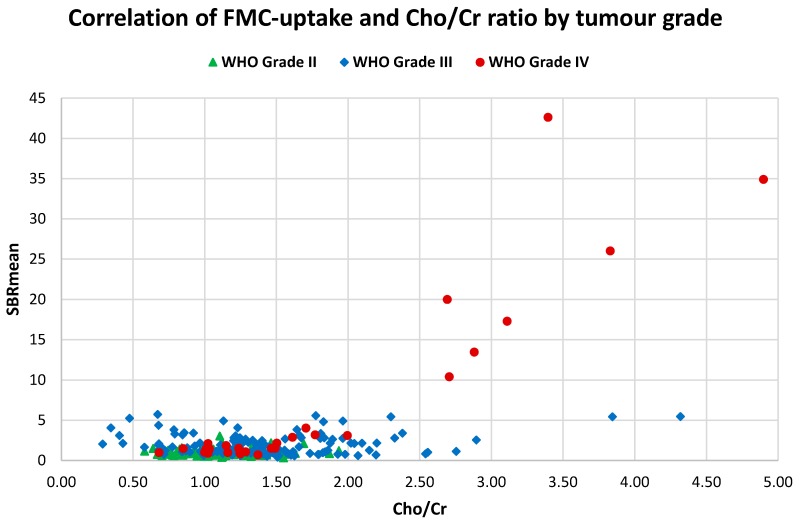
18F-FMC uptake vs. Cho/Cr ratio. A plot of mean sample-to-background (SBR_mean_) against Cho/Cr ratio showed a moderate significant positive correlation between the two measures (r = 0.59, *p* < 0.001), and a strong significant positive correlation when considering WHO grade IV tumours alone (r = 0.9, *p* < 0.001). All data points with an SBR over 10 were from a single patient. Excluding that patient resulted in a weak significant positive correlation between 18F-FMC-uptake and Cho/Cr (r = 0.24, *p* < 0.001).

**Figure 4 cancers-11-01969-f004:**
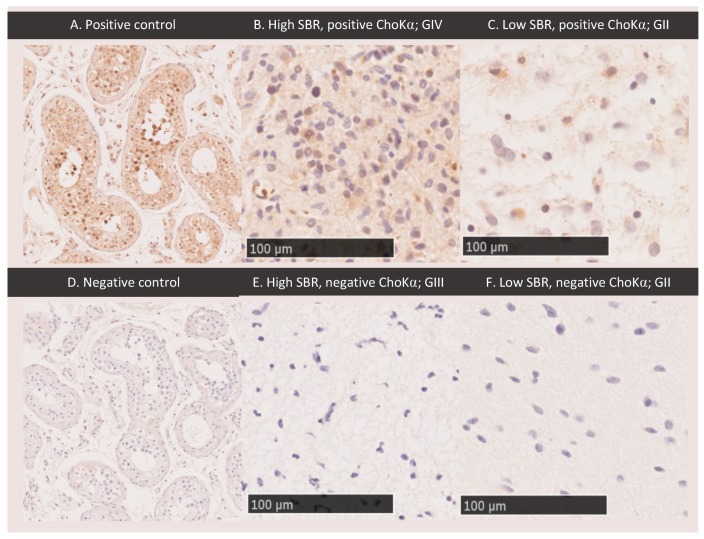
Choline kinase alpha staining. Positive (**A**, ×10) and negative (**D**, ×10) control staining for ChoKα in testes are shown. Glioma tumour tissue samples exhibiting regions of high PET sample-to-background ratio (SBR) (**B**: SBR = 42.6; E: SBR = 3.1) showed both positive and negative ChoKα staining. Glioma tumour tissue samples exhibiting regions of low SBR (**C**: SBR = 1.09; F: SBR = 1.6) showed both positive and negative ChoKα staining. Samples shown are from four different patients: GBM (B, 20×), WHO grade II Astrocytoma (C, 20×), WHO grade III Anaplastic Astrocytoma (**E**, 20×), WHO grade II Astrocytoma (**F**, 20×). MRS Cho/Cr was high for sample B (2.88), and low for samples C (1.23), E (1.89), and F (1.05).

**Figure 5 cancers-11-01969-f005:**
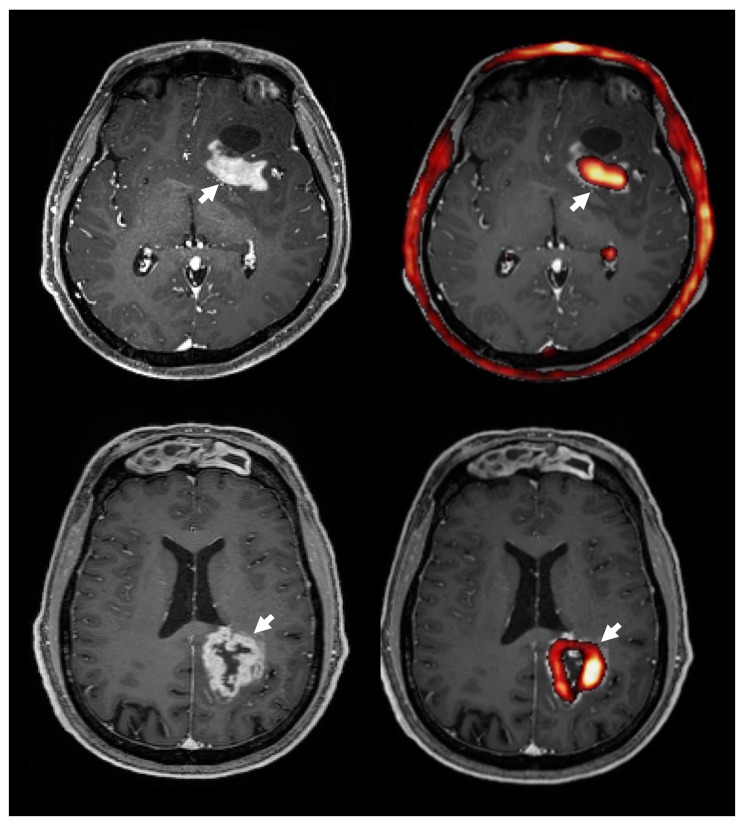
The 18F-FMC uptake (right, red) and contrast enhancement on MRI (left) in two patients with GBM. Overlaying 18F-FMC PET onto post-contrast T1 weighted MRI showed that areas of contrast enhancement had a high 18F-FMC-uptake (white arrows). Overall, an average 94% of contrast enhancing regions had a 18F-FMC-uptake above the threshold, which was defined as three times that in the contralateral white matter. Note the normal high uptake in the choroid plexus and bone marrow.

**Table 1 cancers-11-01969-t001:** Summary of patient tissue data and corresponding PET and MRS results from each sample. M = mutant, WT = wild type, R = retained, CD = codeleted, U = unknown, ChoKα = negative, DNET = diffuse neuroepithelial tumour. Patient 6 (P006) did not have a PET scan, and the location of the surgical resection was not recorded, excluding them from tissue vs. imaging analyses. P007 did not undergo surgical resection at our institution and hence no tissue data were available. P013 (DNET) was excluded from the analyses. Grey cells: data unavailable.

Subject	Tumour Type, WHO Grade	Molecular			Sample 1			Sample 2		
		IDH1	1p/19q	ATRX	SBR (PET)	Cho/Cr (MRS)	ChoKα	SBR (PET)	Cho/Cr (MRS)	ChoKα
P001	Astrocytoma, II	M	R	M	1.6	1.05	-	1	1.86	-
P002	Oligodendroglioma, III	M	U	WT	3.4	1.8	-	2.2	0.42	-
P003	Oligodendroglioma, III	M	CD	WT	4.4	1.65	-	3.6	1.68	weak
P004	Astrocytoma, III	WT	R	M	0.8	0.9	-	0.8	1.59	-
P005	Oligodendroglioma, II	M	R	M	1	0.96	-	1.1	1.23	weak
P006	Oligodendroglioma, II	M	U	WT	unrecorded location		weak	unrecorded location		weak
P007	Astrocytoma, II	surgery at external institution								
P008	GBM	M	R	M	5.6	1.02	weak	no second sample		
P009	GBM	WT	R	M	9.3	N/A	weak	7.3	1.77	weak
P010	GBM	WT	R	U	43	2.88	strong	20	2.7	-
P011	Astrocytoma, III	M	R	M	3.1	1.89	-	3.9	2.91	weak
P012	Astrocytoma, II	M	R	M	1.3	1.53	-	1.2	0.75	weak
P013	DNET, I	excluded from analyses								
P014	Astrocytoma, II	M	R	M	2.7	0.84	-	no second sample		

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
