# Peer review of "Imaging and Tissue Biomarkers of Choline Metabolism in Diffuse Adult Glioma: 18F-Fluoromethylcholine PET/CT, Magnetic Resonance Spectroscopy, and Choline Kinase α"

_cancers, 2019, doi:10.3390/cancers11121969_

Round 1
Reviewer 1 Report
Comments to Authors:
General Comments:
In this manuscript, Grech-Sollars et. al. built upon their 2017 manuscript using PET and MR spectroscopy as diagnostic biomarkers for glioma. The correlation of imaging results with tissue analysis of choline kinase alpha is timely and innovative. This is an important study. However, there are several items that need to be addressed before publication.
Major revisions:
General comments: Based on the author’s previous publication (Reference 17), it is suspected that the patient data used in this current study were previously recruited and used in the prior manuscript. The authors should clearly state this.
Methods:
1. General comment: Patient 13 (DNET) should be excluded from all analyses and reporting of ‘n’. While it is stated that the patient was excluded from analysis, n=14 is often stated and references to patient 13 are frequently made. Please correct this throughout the manuscript.
2. 2.2: 1) How were the positions for the voxels chosen for MRS? 2) Line 103: why was the injection halted after 38 minutes? Was there an adverse event?
3. 2.5.4: Please provide justification for 1) multiplication of 9/3 for conversion of relative metabolite concentration to peak intensity and 2) selection of Cho/Cr threshold of 2.4. 3) Your spectroscopy data should be compared to contralateral brain in all patients. Please show this data.
4. 2.5.5: 1) A t-test to compare the difference in SUVmax, TBR and MRS data across tumor grades is not appropriate. If data passed normality, ANOVA with appropriate post-hoc analysis should be performed to compare all groups. 2) Table 1 indicates that patients 6 and 7 have no PET or MRS data recorded. Patient 13 should be excluded from the analysis. Thus, as written, you have n=11 for PET and MRS data.
Results:
(1) Table 1: It would be much easier to read if the tumor type/grades were grouped. Patient 13 should be removed from this table. Do you not have SBR, MRS and ChoKalpha data for Patient 6 and 7? The way the manuscript reads is that you have PET data for 1 and MRS data for the other. Can you make this more clear on this table?
(2) Figure 1A: Can you indicate the tumor grade with the patient ID in the figure legend for reader convenience?
(3) Figure 1B: Patient 13 should be removed from this analysis.
(4) 3.4 and Figure 2A, 2B: Please reflect the statistical results per ANOVA analysis. Representative PET and MRS images would be great to include.
(5) Figure 4: Please provide scale bar. The images appear to be taken at different magnification. It would be helpful for the reader to have the tumor types indicated in the figure, not just the figure legend.
(6) Consider doing immunostaining for phosphocholine, as this may be better correlated with your imaging data. Also, the tumor microenvironment is very heterogeneous. Your staining is suggestive of non-tumor cell (i.e., infiltrating immune cells) immunopositivity. Double-labeling for microglia/macrophages would be of interest.
(7) 3.8: Again, remove patient 13 from this analysis (WHO I); Line 336. It would be helpful to present these images as a panel: a) T1 post-contrast images, b) PET and C) merge for each patient.
Author Response
General Comments:
In this manuscript, Grech-Sollars et. al. built upon their 2017 manuscript using PET and MR spectroscopy as diagnostic biomarkers for glioma. The correlation of imaging results with tissue analysis of choline kinase alpha is timely and innovative. This is an important study. However, there are several items that need to be addressed before publication. Major revisions: General comments: Based on the author’s previous publication (Reference 17), it is suspected that the patient data used in this current study were previously recruited and used in the prior manuscript. The authors should clearly state this.
Response: We thank the reviewer for this comment. The prior paper (ref 17) focused on the methodology of integrated PET and MRS neuronavigational tool; it did not report on the results of patient data, which has not been previously published. We have amended this in the manuscript. Lines 112-114 now read: “Regions of both high and low Cho/Cr ratios on MRS and 18F-FMC uptake on PET were marked for biopsy by neurosurgeons using color coded hollow virtual spheres represented in the surgical neuronavigational system usingmethodology described previously 17.”
Methods:
General comment: Patient 13 (DNET) should be excluded from all analyses and reporting of ‘n’.While it is stated that the patient was excluded from analysis, n=14 is often stated and references topatient 13 are frequently made. Please correct this throughout the manuscript.Response: Patient 13 was excluded from tissue and MRI analysis. However, PET results for this patient are relevant to one of the conclusions of the study. This patient had enhancing tumour components within which we observed high choline uptake on PET, despite being an indolent lesion of negligible malignant potential. This finding supported our observations in the glioma patients that suggested that tracer uptake could be confounded by blood-brain barrier disruption.
2.2: 1) How were the positions for the voxels chosen for MRS? 2) Line 103: why was the injectionhalted after 38 minutes? Was there an adverse event?Response: Regarding point 1, as stated in 2.2 we selected between one to three slices to cover the whole tumour where possible. Regarding point 2, the patient asked to stop the scan after 38 minutes and hence the acquisition was halted. This was not due to an adverse event. We have now added to the manuscript in line 104: “in another patient the acquisition was halted after 38 minutes as the patient needed to micturate.”
3. 2.5.4: Please provide justification for 1) multiplication of 9/3 for conversion of relative metabolite concentration to peak intensity and 2) selection of Cho/Cr threshold of 2.4. 3) Your spectroscopy data should be compared to contralateral brain in all patients. Please show this data.
Response: Regarding point 1, we chose to report peak intensity in line with the majority of literature, so that our values can be compared to previously published values, e.g. (Usinskiene et al., 2016). These differences reflect the number of protons contributing to each resonance, and the values calculated differ using different MRS analysis programmes– eg TARQUIN vs LC-model.
Regarding point 2, a pragmatic threshold of 2.4 was chosen for comparison with ChoKα expression in spatially matched samples. Amended in Section 2.5.4 to read: ‘Relative values were used to select high and low Cho/Cr regions within individual tumours. A pragmatic Cho/Cr threshold of 2.4 was used to define regions of high and low Cho/Cr for the purposes of comparison with ChoKα staining.’
We have now also acknowledged the limitation of using relative values for selecting tissue targets in section 4.6: ‘Limitations for the correlation between imaging and tissue biomarkers include the limited spatial resolution of the imaging techniques, relative imaging parameters applied for selecting biopsy location, and accuracy of the neuro-navigational system.’
Regarding point 3, our CSI acquisition aimed to cover the whole tumour and we did not acquire MRS data in contralateral brain. We used Creatine as a reference rather than contralateral values, as in many published MRS studies on brain tumours, e.g. (Usinskiene et al., 2016).
2.5.5: 1) A t-test to compare the difference in SUVmax, TBR and MRS data across tumor gradesis not appropriate. If data passed normality, ANOVA with appropriate post-hoc analysis should beperformed to compare all groups. 2) Table 1 indicates that patients 6 and 7 have no PET or MRS data recorded. Patient 13 should be excluded from the analysis. Thus, as written, you have n=11 for PETand MRS data.Response: Point 1: We thank the author for pointing this out. The data did not pass the Shapiro Wilk normality test and hence the Wilcoxon rank-sum test was used to compare differences. This has now been amended in the text. Results have not changed.
Point 2: We thank the author for pointing this out, and have amended our manuscript to say that PET data was analysed in 12 patients and MRS data in 13 patients. The DNET was excluded from the analysis for both. Only patient 6 did not have a PET acquisition. Patient 7 had both PET and MRS but did not receive surgery at our institution, and hence tissue data was not available for this patient.
Results:
(1) Table 1: It would be much easier to read if the tumor type/grades were grouped. Patient 13 should be removed from this table. Do you not have SBR, MRS and ChoKalpha data for Patient 6 and 7? The way the manuscript reads is that you have PET data for 1 and MRS data for the other. Can you make this more clear on this table?
Response: Table 1 shows the results for the analyses within the samples collected. We include patient 13 for completeness, indicating that we excluded this patient from the analyses. Patient 6 did not have a PET scan and in addition, the location of the tissue samples was not recorded during surgery. Hence, we do not report MRS data for Patient 6 in this table. Patient 7 did not undergo surgical resection at our institution and hence we were unable to carry out the tissue analyses. The caption for Table 1 has been amended to: “P006 did not have a PET scan, and the location of the surgical resection was not recorded, excluding them from tissue vs imaging analyses. P007 did not undergo surgical resection at our institution and hence no tissue data was available. P013 (DNET) was excluded from the analyses. ”
(2) Figure 1A: Can you indicate the tumor grade with the patient ID in the figure legend for reader convenience?
Response: This has now been amended.
(3) Figure 1B: Patient 13 should be removed from this analysis.
Response: Figure 1B shows the high tracer uptake observed in P013, which we observed to be in areas of blood brain barrier disruption as reflected in pathological contrast enhancement on MRI. We believe it is important to show this data for P013, as it reiterates the important finding that blood brain barrier disruption has a large effect on choline uptake on PET.
(4) 3.4 and Figure 2A, 2B: Please reflect the statistical results per ANOVA analysis. Representative PET and MRS images would be great to include.
Response: This has now been updated to the Wilcoxon rank-sum test. We feel that images of PET and MRS across each of the grades might make the figure too busy and are therefore opting to not add these in here.
(5) Figure 4: Please provide scale bar. The images appear to be taken at different magnification. It would be helpful for the reader to have the tumor types indicated in the figure, not just the figure legend.
Response: A scale bar has now been added to the figure. We have also added the tumour grade.
(6) Consider doing immunostaining for phosphocholine, as this may be better correlated with your imaging data. Also, the tumor microenvironment is very heterogeneous. Your staining is suggestive of non-tumor cell (i.e., infiltrating immune cells) immunopositivity. Double-labeling for microglia/macrophages would be of interest.
Response: Thank you for this valid comment. The rationale of ChoKα was as the first unique step in phosphorylating choline towards membrane synthesis. Unfortunately, we underestimated how weak the staining would be in glioma despite the evidence of clear expression in control tissue, as highlighted in section 4.5 of discussion. The suggestion for carrying out immunostaining for phosphocholine is reasonable, however it addresses the intermediary metabolite rather than the enzyme that determines flux. Phosphocholine is a small molecule, unlikely to survive formalin fixation, dehydration in alcohols and xylene and paraffin embedding. Unfortunately, no snap frozen tissue is available for the cases included in this study. We have also considered labelling for microglia/macrophages but this is beyond the scope of the present study. We have added your valid point to the discussion section 4.5: ‘In addition, the contribution of non-tumor cells such as infiltrating immune cells could be addressed in future studies by labelling for microglia/ macrophages.’
Finally, co-localisation of anti-ChoKα with markers for microglia and macrophages is likely to be challenging given the low expression of ChoKα in our samples.
(7) 3.8: Again, remove patient 13 from this analysis (WHO I); Line 336. It would be helpful to present these images as a panel: a) T1 post-contrast images, b) PET and C) merge for each patient.
Response: As discussed above, P013 further confirms our finding that choline PET uptake may be related to blood brain barrier disruption. We therefore believe it is in this case important to include this data in the paper to highlight this. We have updated the figure to also show the contrast enhancing regions separately from the PET overlay on the MRI.
References
Usinskiene, J., Ulyte, A., Bjørnerud, A., Venius, J., et al. (2016) Optimal differentiation of high-and low grade glioma and metastasis: a meta-analysis of perfusion, diffusion, and spectroscopy metrics. Neuroradiology. 58 (4), 339–Available from: doi:10.1007/s00234-016-1642-9.
Reviewer 2 Report
Though the sample size is limited, it is a nice clinic research article. Suggest to accept it.
Author Response
We would like to thank the reviewer for their positive feedback.
Reviewer 3 Report
The authors shoed that the relationship between choline radiotracer uptake and MRS-visible metabolites by using 18F-FMC PET/CT, MRS and tissue choline kinase alpha (ChoKα) staining. The authors demonstrated the advantages of 18F-FMC PET and MRS to provide the complimentary information on glioma grade, since ChoKα expression is generally low. I wonder why the authors chose ChoKα as a marker for the immunostaining despite its generally low expression in diffuse glioma tissue. The appropriate markers for immunostaining should be used to assess the association of PET uptake and MRS with tissue markers for glioma grade. Further concerns are detailed below.
In some experiments, the authors excluded some data (e.g. line 239, 280). The exclusion rule should be clearly determined and mentioned in Method section.
Fig. 2B. the authors detected the significant difference in [Cho/Cr]max between WHO grade II and III. The significance was not depicted with asterisk as in Fig. 2A.
Table 1 and Fig. 4. How did the authors determine the staining intensity (weak or strong)? Generally, paraffin-embedded tissue for immunohistochemistry is “semi-quantitative”. Other quantitative and statistical analysis should be required to support the author’s claim. As above mentioned, the other glioma markers such as Ki67 should be assessed for immunostaining.
Author Response
The authors shoed that the relationship between choline radiotracer uptake and MRS visible metabolites by using 18F-FMC PET/CT, MRS and tissue choline kinase alpha (ChoKα) staining. The authors demonstrated the advantages of 18F-FMC PET and MRS to provide the complimentary information on glioma grade, since ChoKα expression is generally low. I wonder why the authors chose ChoKα as a marker for the immunostaining despite its generally low expression in diffuse glioma tissue. The appropriate markers for immunostaining should be used to assess the association of PET uptake and MRS with tissue markers for glioma grade. Further concerns are detailed below.
Response: Thanks for raising this, as stated in the reply to Reviewer 1, comment 6, the rationale of ChoKα was as the first unique step in phosphorylating choline towards membrane synthesis. There are limited published data on in vivo ChoKα expression in glioma. Unfortunately, we underestimated how weak the staining would be in glioma despite the evidence of clear expression in control tissue, as highlighted in section 4.5 of discussion. We have added to section 4.5 of the discussion 'In addition the contribution of non-tumor cells such as infiltrating immune cells could be addressed in future studies by labelling for microglia/ macrophages.'
In some experiments, the authors excluded some data (e.g. line 239, 280). The exclusion rule should be clearly determined and mentioned in Method section.
Response: The exclusion is mentioned in the Method in Section 2.1 and was excluded on the basis that it is not a glioma.
Fig. 2B. the authors detected the significant difference in [Cho/Cr]max between WHO grade II and III. The significance was not depicted with asterisk as in Fig. 2A.
Response: We have not used any asterisk to mark significance in either figure 2A or 2B. What might be seen by the reviewer as an asterisk on the NAWM is one of the data points for one of the patients which was higher than all the other points and marked as an outlier.
Table 1 and Fig. 4. How did the authors determine the staining intensity (weak or strong)? Generally, paraffin-embedded tissue for immunohistochemistry is “semi-quantitative”. Other quantitative and statistical analysis should be required to support the author’s claim. As above mentioned, the other glioma markers such as Ki67 should be assessed for immunostaining.
Response: We agree that quantification is important when correlating findings. However, in all weak samples, positive cells were very few and quantification would not have given additional information to the lack of correlation that we already observed in determining positive or negative staining. Regardless of quantification, as shown in Table 1, there was no correlation between imaging and positive/negative ChoKα: regions with high and low Cho/Cr on MRS stained both positive and negative; and regions with high and low SBR on PET stained both positive and negative. Ki67 was not assessed as it would not help to test the hypothesis of whether choline metabolism as measured on imaging is related to ChoKα.
Round 2
Reviewer 1 Report
Your changes are acceptable. Thank you for addressing my concerns.
Reviewer 3 Report
The authors addressed all of my concerns.